# Effects of *Lactococcus cremoris* PS133 in 5-Hydroxytryptophan-Induced Irritable Bowel Syndrome Model Rats

**DOI:** 10.3390/ijms26062464

**Published:** 2025-03-10

**Authors:** Fu-Sheng Deng, Miao-Hui Lin, Chin-Lin Huang, Chien-Chen Wu, Ching-Liang Lu, Ying-Chieh Tsai

**Affiliations:** 1Bened Biomedical Co., Ltd., Taipei 115011, Taiwan; stephen@benedbiomed.com (F.-S.D.); hjenny@benedbiomed.com (C.-L.H.); 2Institute of Biochemistry and Molecular Biology, National Yang Ming Chiao Tung University, Taipei 11221, Taiwan; a50253@hotmail.com.tw; 3Biomedical Industry Ph.D. Program, National Yang Ming Chiao Tung University, Taipei 11221, Taiwan; 4School of Medicine, National Yang Ming Chiao Tung University, Taipei 11221, Taiwan; cllu@vghtpe.gov.tw; 5Institute of Brain Science, National Yang Ming Chiao Tung University, Taipei 11221, Taiwan; 6Endoscopy Center of Diagnosis and Treatment, Taipei Veterans General Hospital, Taipei 11221, Taiwan; 7Division of Gastroenterology, Taipei Veterans General Hospital, Taipei 11221, Taiwan

**Keywords:** *Lactococcus cremoris*, PS133, irritable bowel syndrome, gut microbiota

## Abstract

Irritable bowel syndrome (IBS) is a functional gastrointestinal disorder marked by abdominal pain and irregular bowel habits. Recently, more and more evidence supports gut microbiota imbalance in IBS and highlights the potential of probiotics in restoring gut health and reducing symptoms. In this study, we explored the effects of *Lactococcus cremoris* PS133 (PS133) on an IBS-like condition in rats triggered by 5-hydroxytryptophan (5-HTP), a serotonin precursor. Eight-week-old Sprague Dawley rats received either PS133 or saline for 14 days, followed by 5-HTP to induce IBS-like symptoms. Colorectal distension tests showed that PS133 reduced visceral hypersensitivity. PS133 also protected intestinal mucin against 5-HTP-induced degradation, as seen in alcian blue staining, and increased the levels of tight junction proteins (occludin and zonula occludens-1) in the colon, indicating improved gut barrier integrity. Additionally, PS133 normalized the levels of substance P (a neuropeptide) in the spinal cord and altered 5-hydroxyindoleacetic acid (a serotonin metabolite) in the brain. Gut microbiota analysis revealed PS133 regulated specific bacterial groups, including [*Eubacterium*]*_coprostanoligenes_group* and *Lactococcus*. Overall, PS133 improved gut function, reduced IBS-like symptoms, and modulated gut microbiota, neurotransmitters, and intestinal barrier health in this IBS model.

## 1. Introduction

Irritable bowel syndrome (IBS) is a chronic disorder that affects millions of individuals globally. According to statistics, IBS accounts for approximately 11% of cases in the general population with a slight female predominance [1]. In general, IBS is not considered a serious threat; however, it is commonly associated with functional gastrointestinal disorders, including abdominal discomfort, altered bowel habits (constipation or diarrhea), and other functional digestive disorders [2]. Although IBS is not associated with increased mortality, it still imposes a significant burden on patients, healthcare systems, and society [3]. In the United States, the total annual cost of IBS is more than $20 billion [4].

To date, the exact etiology of IBS remains unclear. Multiple factors share a correlation with IBS, including visceral hypersensitivity (VH), dysmotility, psychological disturbances, food intolerance, gut dysbiosis, genetic factors, and infection exposure [4,5,6]. Among these, VH is one of the primary characteristics of patients with IBS [7]. Furthermore, over 30% of patients with IBS are more sensitive to colonic dilatation than healthy controls [8]. Neurotransmitters and their receptors may also play a role in VH [9]. 5-hydroxytryptamine (5-HT), or serotonin, is a monoamine neurotransmitter that is metabolized to 5-hydroxyindoleacetic acid (5-HIAA). Evidence suggests that altered 5-HT signaling may alter intestinal motility, boost VH, increase intestinal permeability, and induce inflammation in patients with IBS [10]. For example, 5-hydroxytryptophan (5-HTP), a 5-HT precursor, is commonly used to induce IBS-like symptoms in animal models. Scientific evidence has revealed that 5-HTP injection activates distal colonic myenteric neurons via dose-dependent 5-HT_4_ receptors, increases fecal fluid content, and induces colonic motor function [11]. Based on this 5-HT alteration theory, various treatment approaches have been suggested. 5-HT receptor antagonists such as alosetron, cilansetron, and tegaserod are effective in treating IBS; however, they may cause ischaemic colitis and cardiovascular events as side effects [12]. Furthermore, some psychotherapies or antidepressants have been prescribed for treating patients with IBS; however, their efficacy is unsatisfactory [13]. Currently, safe and effective therapies are scarce.

According to the World Health Organization, probiotics are defined as ‘live microorganisms which, when administered in adequate amounts, confer a health benefit on the host’ [14]. When ingested in adequate amounts, specific probiotics modulate gut microbiota composition and regulate brain function and behavioral performance via several pathways, including neural, immune, endocrine, and microbiota-driven metabolites, which constitute the microbiota–gut–brain axis (MGBA) [15,16]. By modulating the MGBA, regulating the mucosal immune system, and improving intestinal barrier function, probiotic supplementation may affect IBS development [17]. For example, *Lactobacillus plantarum* PS128 (PS128) is a probiotic strain that attenuates 5-HTP-induced VH by modulating the MGBA [18]. Furthermore, short-term supplementation of active lactic acid bacteria exhibits effectiveness and safety for improving intestinal mucosal barrier function in patients with IBS by upregulating the tight junction proteins occludin and zonula occludens-1 (ZO-1) [19]. Many experimental and clinical studies have revealed the positive effects of *Bifidobacterium infantis* 35624 on immune responses and IBS symptoms [20,21]. Similarly, *Lactobacillus paracasei* NCC2461, *Lactobacillus farciminis* CIP103136, and *Lactobacillus rhamnosus* LCR35 have exhibited effectiveness in IBS rodent models [22,23,24]. Although probiotics have demonstrated preliminary efficacy in IBS treatment, the exact mechanisms underlying these interactions remain unclear [9].

*Lactococcus cremoris* PS133 (PS133), isolated from Taiwanese fermented cabbage, has been shown to exhibit immunomodulatory effects in mice [25]. In the present study, we investigated whether PS133 affects 5-HTP-induced IBS model rats. Furthermore, we evaluated its impact on pain-associated neurotransmitter levels, gut barrier function, and gut microbiota composition. Our findings provide comprehensive insights into the role of *L. cremoris* PS133 in alleviating IBS-related symptoms.

## 2. Results

### 2.1. Effect of L. cremoris PS133 on 5-HTP-Induced VH

As previously described, VH changes observed during 5-HTP model establishment can be utilized as an IBS severity metric and can be measured using CRD [18]. Herein, we performed CRD to evaluate the effects of PS133 in rats with 5-HTP-induced VH. Before PS133 administration, the visceromotor reflex (VMR) levels were significantly increased with 5-HTP injection but comparable between the Saline and PS133 groups (Appendix A). After 14 days of oral supplementation, the baseline VMR levels were still comparable between the Saline and PS133 groups (Figure 1A,B, baselines). Interestingly, 5-HTP injection still significantly increased the VMR levels at three distension pressures compared with baseline levels in the Saline group (*p* < 0.01 at 30 mmHg and *p* < 0.001 at 60 and 80 mmHg) (Figure 1A) but not in the PS133 group (Figure 1B). Collectively, these results suggest that *L. cremoris* PS133 ameliorates 5-HTP-induced VH in rats.

### 2.2. Effect of L. cremoris PS133 on Substance P (SP) Levels

SP is a neurotransmitter associated with the transmission of pain information to the spinal cord and brain [26]. A previous study revealed that visceral pain signals in CRD-treated mice are primarily transmitted to the L6-S1 segments of the spinal cord [27]. To further confirm the effect of PS133 on 5-HTP-induced VH, we performed immunofluorescence staining to measure SP levels in the L6-S1 segments of the spinal cord. Figure 2A illustrates SP immunoreactivity in the L6 and S1 segments in all four groups. In the L6 segment, weaker SP immunoreactivity was observed in the Naïve and Sham groups; however, stronger SP immunoreactivity was observed in the Saline group (Figure 2B). Notably, the intensity of SP-immunoreactive fibers in the L6 spinal cord was significantly lower in the PS133 group than in the Saline group (Figure 2B, *p* < 0.05). In the S1 segment, we did not observe stronger SP intensity in the Saline group. A similar result was noted in the S1 segment: PS133 administration decreased 5-HTP-induced SP immunoreactivity, with the intensity even lower than in rats in the Naïve group (Figure 2C, *p* < 0.05). Collectively, these observations suggest that *L. cremoris* PS133 substantially decreases SP levels in rats with 5-HTP-induced VH.

### 2.3. Effect of L. cremoris PS133 on Mucin Expression

Mucins are major components of the colonic mucus barrier [20,28]. To investigate whether PS133 supplementation affects mucin expression in the rat model of 5-HTP-induced IBS, we performed alcian blue staining to analyze mucin production in the distal colon of rats. Mucins were detected on the surface of the colonic epithelium. The mucin-containing areas were stained, counted, and normalized to the positive area of the tissue (Figure 3A). Figure 3B illustrates the stained areas and reveals that compared with the Naïve group, the colonic mucin amounts were significantly decreased in three CRD-treated groups, suggesting CRD causes mucosal damage in the gut. Accordingly, a smaller positive colonic mucin area was observed in the Saline group compared with the Sham group. This suggests that 5-HTP injection damages the intestinal mucosal barrier. Interestingly, PS133 supplementation markedly restored the 5-HTP-induced changes in colonic mucin expression. Furthermore, the mucin-containing area was significantly larger in the PS133 group than in the Sham group (Figure 3B, *p* < 0.05).

### 2.4. 5-HTP-Induced Changes in Tight Junction Protein Levels and Effect of L. cremoris PS133 Supplementation

A previous study revealed a positive correlation between increased intestinal epithelial permeability and VH in patients with IBS [29]. Tight junction proteins such as occludin and ZO-1 maintain the intestinal barrier and regulate ion, nutrient, and water permeability [30]. To assess intestinal permeability, we analyzed the intensities of occludin and ZO-1 in the colonic mucosa using immunofluorescence staining. As shown in Figure 4A,B, the network of occludin (red) and ZO-1 (green) was predominantly intact and localized along the apical cellular border in both the Naïve and Sham groups. The Sham group even exhibited higher intensities of occludin and ZO-1 than the Naïve group (Figure 4C,D), suggesting that CRD treatment stimulates tight junction protein expression. However, compared with the Sham group, the intensities of the occludin and ZO-1 proteins were significantly lower in the Saline group (Figure 4C,D, *p* < 0.05). Importantly, 14 days of PS133 supplementation elevated occludin and ZO-1 levels in 5-HTP-induced IBS model rats. Figure 4C,D illustrate that the intensities of the occludin and ZO-1 proteins were significantly higher in the PS133 group than in the Saline group, which was not supplemented with PS133. Collectively, these data suggest that *L. cremoris* PS133 supplementation promotes the appropriate expression and distribution of tight junction proteins.

### 2.5. Effects of L. cremoris PS133 on Neurotransmitter Expression in the Brains of 5-HTP-Induced IBS Model Rats

5-HTP is a precursor of 5-HT and 5-HIAA. As previously reported, 5-HTP injection significantly affects the 5-HIAA/5-HT ratio in the prefrontal cortex and hypothalamus; interestingly, probiotic supplementation has been shown to restore this balance [18]. Therefore, to investigate the effects of PS133, we harvested the prefrontal cortex and hypothalamus and measured the 5-HT and 5-HIAA levels via HPLC. In the prefrontal cortex, 5-HT and 5-HIAA levels were comparable in the Naïve and Sham groups; this suggests that CRD does not alter 5-HT or 5-HIAA levels (Figure 5A,B). However, after 5-HTP injection, 5-HT and 5-HIAA levels significantly increased in the Saline group (Figure 5A,B, *p* < 0.05). In contrast, 5-HT and 5-HIAA levels, even the 5-HIAA/5-HT ratio, were slightly reversed in the PS133 group with no statistically significant difference (Figure 5A–C). In the hypothalamus, no significant differences were observed in 5-HT levels among the four groups (Figure 5D). However, in the Saline group, the 5-HIAA level was higher in the hypothalamus, significantly increasing the 5-HIAA/5-HT ratio (Figure 5E,F, *p* < 0.05). After *L. cremoris* PS133 supplementation, the 5-HIAA level was reversed in the PS133 group; however, the difference was not statistically significant (Figure 5E). Therefore, the 5-HIAA/5-HT ratio was significantly higher in the hypothalamus of the Saline group than in that of the Naïve and Sham groups (Figure 5F, *p* < 0.05).

### 2.6. Effect of 5-HTP Induction and L. cremoris PS133 on the Gut Microbiota Profile

To investigate the effects of 5-HTP induction and PS133 supplementation on the rat gut microbiota, we extracted DNA from rat cecal contents and performed 16S rRNA amplicon (V3-V4) paired-end sequencing. Microbiota composition analysis revealed *Bacteroidetes* and *Firmicutes* as the dominant phyla among the groups (Figure 6A). Next, we elucidated the richness and evenness of the gut microbiota using the Chao1 index, Shannon’s diversity index, and Simpson’s evenness; however, no significant differences were observed among the groups (Figure 6B–D). To further estimate the β-diversity among the four groups, NMDS was performed, followed by an analysis of similarities (ANOSIM and ADONIS). Figure 6E illustrates the distinct clustering of the gut microbiota composition among the groups (ANOSIM: R = 0.216, *p* = 0.001; ADONIS: R^2^ = 0.164, *p* = 0.005). To further analyze the effects of PS133, linear discriminant analysis effect size was performed to identify the differentially presented taxa and determine the differences in the relative abundances of the microbes between the Saline and PS133 groups. At the genus level, the relative abundance of two genera, namely, *Lactococcus* and [*Eubacterium*]*_coprostanoligenes_group*, was associated with *L. cremoris* PS133 supplementation in rats with 5-HTP-induced IBS (Figure 6F). Interestingly, an increasing trend in the abundance of [*Eubacterium*]*_coprostanoligenes_group* was also observed in another IBS model [31]. Then, we measured the abundance of *Lactococcus* and [*Eubacterium*]*_coprostanoligenes_group* in the four experimental groups. Figure 6G demonstrates that the abundance of *Lactococcus* significantly increased after PS133 supplementation. In addition, the abundance of [*Eubacterium*]*_coprostanoligenes_group* was significantly increased in the Sham and Saline groups compared with the Naïve group; importantly, it was restored in the PS133 group (Figure 6H). Collectively, these results demonstrate that 5-HTP injection followed by CRD alters the gut microbiota composition and that the composition is modulated by *L. cremoris* PS133.

## 3. Discussion

IBS is a functional bowel disorder characterized by symptoms such as abdominal pain or discomfort; this condition can negatively affect the quality of life of patients [32,33]. Therefore, developing safe and effective therapies for IBS, including probiotic intake, which exerts beneficial effects, is encouraged. In this study, we elucidated the effects of *L. cremoris* PS133 on 5-HTP-induced VH and investigated gut barrier function and the gut microbiota to explore the underlying mechanisms. We observed that PS133 ameliorates 5-HTP-induced IBS-like deficits via the gut–brain axis, as evidenced by a comprehensive analysis of visceromotor reflex performance, SP immunoreactivity in the spinal cord, tight junction protein levels in the colon, neurotransmitter levels in the brain, and gut microbiota composition in rats with IBS-like symptoms.

VH is a characteristic of patients with IBS. Physical stress or psychological anxiety can induce VH. It is strongly associated with neurotransmitter dysfunction [34]. Based on the findings of previous studies, we investigated 5-HIAA and 5-HT levels after 5-HTP injection and focused on the prefrontal cortex and hypothalamus, which are involved in visceral homeostasis and are dysregulated in IBS pathophysiology [35,36]. We observed that 5-HIAA levels were significantly increased in the prefrontal cortex and hypothalamus after 5-HTP injection; in contrast, PS133 supplementation significantly decreased its levels (Figure 5B,E). To date, the effects of probiotics on 5-HIAA metabolism have not been thoroughly investigated in rats with 5-HTP-induced VH. Therefore, additional studies are warranted to elucidate the precise mechanism underlying the inhibitory effect of PS133 on 5-HIAA levels in rats with 5-HTP-induced VH. SP, a pain-mediating neurotransmitter, serves as another vital mediator in VH processing [37,38]. As previously described, PS128 exerts a positive ameliorative effect on VH in 5-HTP-induced IBS model rats [18]. Mechanistically, the ameliorative effects of PS128 on VH are accompanied by decreased serum corticosterone levels, restored neurotransmitter protein levels (including SP and 5-HIAA), and increased glucocorticoid receptor levels but decreased mineralocorticoid receptor levels in the amygdala [18]. In the present study, we demonstrated that PS133 exerts positive effects on VH by restoring the 5-HIAA/5-HT ratio and SP levels in 5-HTP-induced rats; however, serum corticosterone levels were not significantly decreased (Appendix A). Corticosterone, the main biomarker for the degree of stress in rodents, is an essential molecule for determining the stress responses of experimental animals [39]. Recent clinical studies have implicated the dysregulation of the hypothalamic–pituitary–adrenal (HPA) axis in IBS pathophysiology by revealing elevated cortisol levels both at baseline and in response to stress [40,41]. However, we did not observe a significant decrease in corticosterone levels in rats with 5-HTP-induced VH and CRD after PS133 supplementation. Nevertheless, the levels of other stress hormones, including epinephrine and norepinephrine, may ameliorate VH [42,43]. Therefore, in the future, the effects of PS133 on the levels of other stress hormones and the imbalance of the HPA axis in rats with IBS should be comprehensively investigated.

Many studies have revealed that altered microbial composition is associated with changes in the mucosal and systemic levels of microbial metabolites in patients with IBS, further linking the microbiota composition to symptoms [44,45]. In animal models, a recent study revealed that rats with neonatal maternal-separation-induced IBS have a higher abundance of *Alloprevotella*, *Bacteroidales*, [*Eubacterium*]*_coprostanoligenes_group*, *Prevotellaceae_Ga6A1_group*, *Romboutsia*, and *Bifidobacterium* but a lower abundance of *Enterococcaceae* and *Lachnospiraceae_NK4A136_group* than healthy controls [31]. Interestingly, in the present study, we observed that *L. cremoris* PS133 oral supplementation significantly altered the abundance of the gut microbiome in IBS model rats. Compared with the Naïve group, the abundances of *Bacteroidales*, [*Eubacterium*]*_coprostanoligenes_group*, and *Prevotellaceae_Ga6A1_group* were significantly increased, but that of *Lachnospiraceae_NK4A136_group* was significantly decreased in the Saline group (Appendix A). The variation trends for the four genera are largely consistent with the result of the previous study [31]. After 14 days of PS133 supplementation, the abundance of [*Eubacterium*]*_coprostanoligenes_group* was restored (Appendix A), similar to that in the Naïve group (Appendix A). According to the previous study, the abundance of [*Eubacterium*]*_coprostanoligenes_group* is associated with cholesterol levels, with a high relative abundance in IBS model rats [31,46]. Evidence suggests that the abundance of [*Eubacterium*]*_coprostanoligenes_group* is positively correlated with the acetate proportion but negatively correlated with gas production, dry matter digestibility, and propionate levels [47]. Furthermore, *Lachnospiraceae_NK4A136_group* belongs to the *Lachnospiraceae* family, whose members are potent short-chain fatty acid (SCFA) producers. Previous research has revealed that the abundance of *Lachnospiraceae* is significantly decreased in patients with IBS [48]. Collectively, we hypothesized that improved SCFAs or other fecal metabolite levels by regulating the microbial composition could be one of the potential mechanisms by which PS133 improves IBS.

Although we demonstrated that *L. cremoris* PS133 supplementation can modulate the gut microbiota, increase mucin production, improve intestinal permeability, and alleviate 5-HTP-induced VH, our study has some limitations. First, we did not perform metabolomics analysis of rat fecal and blood samples to investigate the mechanism by which altered gut microbiota affects IBS via the MGBA. Second, IBS is associated with a high prevalence of psychological disorders, particularly depression and anxiety [49,50]. However, we did not investigate whether *L. cremoris* PS133 can improve depression or anxiety in rats with IBS-like symptoms. In the future, the microbiota community composition and metabolome of IBS models should be comprehensively analyzed to clarify the mechanism by which probiotics alleviate IBS.

## 4. Materials and Methods

### 4.1. Preparation of L. cremoris PS133

*L. cremoris* PS133 (formerly *Lactococcus lactis* subsp. *cremoris*; DSM 27109) was cultured and prepared as described previously with slight modifications [25]. Cryopreserved PS133 was inoculated in Man Rogosa Sharpe (MRS) broth (Difco Corp., MD, USA), anaerobically cultured at 37 °C for 18 h, and then harvested by centrifugation at 6000× *g* for 10 min. The pellet was resuspended in MRS broth supplemented with 15% glycerol and then stored at −80 °C. Before the oral administration, aliquots of PS133 were thawed in a 37 °C water bath for 1 h and then centrifuged at 6000× *g* for 10 min. The supernatant was removed, and the bacterial pellet was resuspended with 1 mL saline to a final concentration of 5 × 10^10^ colony-forming units per milliliter (CFU/mL).

### 4.2. Animal Preparation

The grouping and schedule for the animal experiments are depicted in Appendix A. Male Sprague Dawley (SD) rats (aged 4 weeks old; weighing 92~98 g) were purchased from BioLASCO Taiwan Co., Ltd. (Taipei, Taiwan). All rats were acclimatized for 14 days under standard conditions in the Laboratory Animal Center of the National Yang Ming Chiao Tung University (NYCU). The room was maintained at a constant temperature (22 ± 1 °C) and humidity (55–65%) with a 12 h light/dark cycle. The rats were fed ad libitum with a standard chow diet and sterilized water. All experiments were performed in accordance with relevant guidelines and regulations and were approved by the Institutional Animal Care and Use Committee of NYCU (IACUC number: 1001102r).

### 4.3. Colorectal Distension (CRD) Procedure with Electromyography (EMG) Recording

To evaluate visceral hypersensitivity in rats, electrodes were implanted using the established protocol [51]. After around 2 weeks of recovery, rats need to be trained for 3 days to the experimental conditions by placing them singly in the tunnel for 3 h per day. The CRD balloon was composed of a latex glove finger (7 cm long) attached to a rectal catheter (Medtronic, Skovlunde, Denmark). The balloon was inflated and left overnight to help equilibrate the tension in its wall. During the CRD testing, conscious rats were restrained in plastic tunnels (6 cm diameter, 25 cm length) and connected to the CRD apparatus and amplifier. The distending balloons were placed in the rectum, 1 cm proximal to the anus, and secured by tape to the tail base. The tube was then connected to a barostat machine (Medtronic, Skovlunde, Denmark). The colon was distended by inflating the balloon to the desired pressure (30, 60, or 80 mmHg) for 10 s intervals with 30 s intervals between distensions (as shown in Appendix A). Distensions were repeated 5 times for each experimental protocol with 5 min intervals between each series to be a session of CRD. To determine the visceromotor response to CRD, the EMG data were collected by a CED 1401 instrument and analyzed with Spike 2 software for Windows (Cambridge Electronic Design, Cambridge, UK). The raw signal was rectified off-line, and the area under the curve for baseline activities in each session was subtracted from the area under the curve for the rectified responses to CRD to obtain the difference between the areas under the curves. In each session, the EMG values from individual distensions were averaged.

### 4.4. Histological Analysis

Rats were deeply anesthetized and perfused with 10% formalin fixative (JT Baker, Center Valley, PA, USA). Colon tissues and the L6-S1 spinal cord were harvested and prepared as described previously. Briefly, colon tissues were fixed in 10% formalin solution, embedded in paraffin, sectioned into 5 µm slices, deparaffinized, and further stained with alcian blue. For immunofluorescence analysis, colon tissue and L6-S1 spinal cord were postfixed with 10% formalin for 4 h at 4 °C and dehydrated thrice with 30% sucrose solution. After drying, samples were totally embedded in OCT gel (OCT compound, SAKURA Finetek, Torrance, CA, USA), frozen at −20 °C for 10 min, and sliced into 10 (for colon) or 30 µm (for spinal cord) thick sections by a cryostat sectioning machine (CM1900, Leica, Wetzlar, Germany). The sections were collected and slowly placed on charged glass slides (saline coating). To remove OCT gel, sections were incubated for 15 min in PBS (pH 7.4) and washed twice with 100 µL TBS containing 0.3% Triton X-100 (TBST). Then, colon sections and L6-S1 spinal cord sections were blocked with 5% goat serum (added 1% BSA) or 5% skim milk for 2 h, respectively; after that, they were incubated overnight with primary antibodies (rabbit anti-ZO-1 antibody, 1:25, Invitrogen, Waltham, MA, USA; rabbit anti-occludin antibody, 1:50, Proteintech, San Diego, CA, USA; rat anti-substance P, 1:200, GeneTex, Irvine, CA, USA) at 4 °C. Subsequently, the sections were washed with 100 µL TBST twice and incubated with secondary antibodies (Goat anti-rabbit-FITC antibody, 1:400, Millipore; Rabbit anti-rat-FITC, 1:400, Jackson ImmunoResearch, West Grove, PA, USA) in the dark for 2 h. Finally, the sections were washed with 100 µL TBST twice, covered with Fluoromount-G™ Mounting Medium (00-4958-02; Invitrogen), and stored at 4 °C. Fluorescent signals were detected using a fluorescence microscope (FV10i, Olympus, Tokyo, Japan), and images were analyzed using the MetaMorph v7.8.0.

### 4.5. Brain Neurotransmitter Analysis

The levels of 5-HT and 5-HIAA in the prefrontal cortex and hypothalamus tissue samples were detected using previously published high-performance liquid chromatography with electrochemical detection (HPLC-ECD) [52]. Briefly, tissue samples of approximately 0.1 g in wet weight were lysed by sonication in the perchloric acid buffer (0.1% HClO_4_, 0.1 mM EDTA, and 0.1 mM Na_2_S_2_O_5_) and then centrifuged at 4 °C with 12,000× *g* for 10 min. The supernatants were filtered with a 0.22 µm membrane filter. The HPLC was equilibrated with the mobile phase (92% of 100 mM NaH_2_PO_4_, 0.74 mM SDS, 0.027 mM EDTA, and 2 mM KCl, pH 3.74, and 8% methanol). The filtered brain tissue samples (20 µL) were injected into the column (Kinetex C18, 2.6 μm, 100 × 2.1 mm I.D.; Phenomenex, Torrance, CA, USA). The concentrations of 5-HT and 5-HIAA were measured by comparing with the standard product plotting curve.

### 4.6. Corticosterone Analysis

The whole blood was collected and allowed to clot on ice. After clotting, serum was prepared by centrifugation (3000× *g*, 10 min, 4 °C) and stored at −80 °C. Serum corticosterone levels were measured using a corticosterone ELISA kit (Cayman chemical, Ann Aribor, MI, USA) according to the manufacturer’s instructions. The sensitivity of the kit was approximately 30 pg/mL.

### 4.7. Gut Microbiota Analysis

We followed the published method to extract bacterial DNA from cecal content [52]. To prepare the 16S rRNA gene amplicons, the V3–V4 region of the bacterial 16S rRNA gene was amplified by PCR with a bacterial universal primer set (341F: 5′-CCTACGGGNGGCWGCAG-3′; 805R: 5′-GACTACHVGGGTATCTAATCC-3′). Then, amplicons were sequenced by Illumina MiSeq (Illumina, San Diego, CA, USA), and 300 bp paired-end reads were generated. The forward and reverse reads of each paired raw read were filtered through DADA2, denoising, and chimera removal to obtain an amplified sequence variant (ASV), and the 16S rRNA reference database was Silva 138. Then, the RDP Classifier Bayez algorithm was used to perform a taxonomic analysis of the representative sequences of ASV, and the community composition of each sample was counted at different species classification levels. In total, 3,348,771 sequences were clustered into 1525 ASV. Based on the distance in the Bray–Curtis matrix, the α-diversity and β-diversity, including, Shannon’s diversity index, the Chao1 index, Simpson’s evenness, and nonmetric multidimensional scaling (NMDS), were analyzed, and dynamic visualization plots were generated using QIIME2 (vision 2019.4.0).

### 4.8. Statistical Analysis

Data were analyzed using GraphPad Prism (version 5) and presented as the mean ± standard error of the mean. A comparison of means among groups was performed using one-way ANOVA with Tukey’s post hoc test, two-way ANOVA with Tukey’s post hoc test, and two-way repeated-measure ANOVA with Bonferroni correction, or Student *t*-test. Statistical significance was set at a two-tailed *p* value of less than 0.05.

## 5. Conclusions

In this study, we observed that *L. cremoris* PS133 supplementation alleviates 5-HTP-induced VH, protects gut epithelial barrier function, and balances SP levels and the 5-HIAA/5-HT ratio in the central nervous system, possibly through the modulation of gut microbiota composition. This highlights its potential as a probiotic agent for ameliorating IBS pathophysiology.

## Figures and Tables

**Figure 1 ijms-26-02464-f001:**
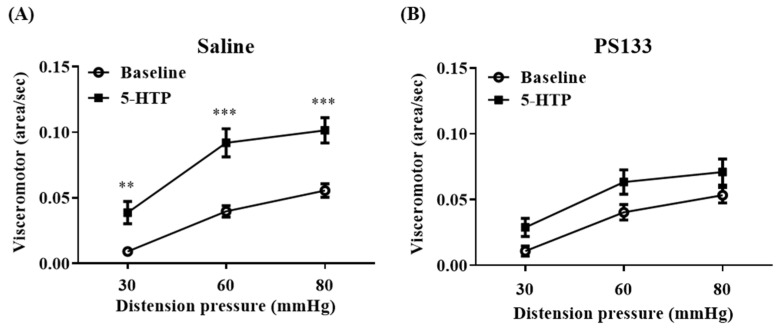
Effect of *L. cremoris* PS133 on 5-HTP-induced VH on experimental day 28. The visceromotor responses were recorded 30 min before (baseline) and after the injection of 5-HTP to (**A**) the Saline group and (**B**) PS133. ** *p* < 0.01; *** *p* < 0.001, compared with the baseline by repeated two-way ANOVA with Bonferroni correction.

**Figure 2 ijms-26-02464-f002:**
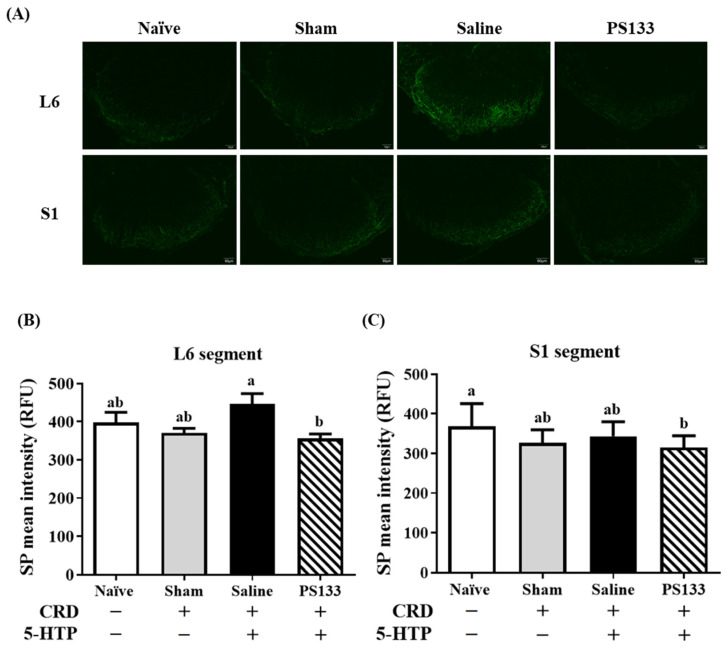
Oral administration of *L. cremoris* PS133 decreased substance P (SP)-immunoreactivity (IR) in L6-S1 spinal cord. SP-IR was detected by (**A**) immunohistochemistry (scale bar = 60 μm) and quantitative analysis of change in intensity of SP in (**B**) L6 and (**C**) S1 segments. All data were means ± S.E.M. and were analyzed by two-way ANOVA with Tukey’s post hoc test. Different letters (a and b) above bars indicate statistically significant differences at *p* < 0.05.

**Figure 3 ijms-26-02464-f003:**
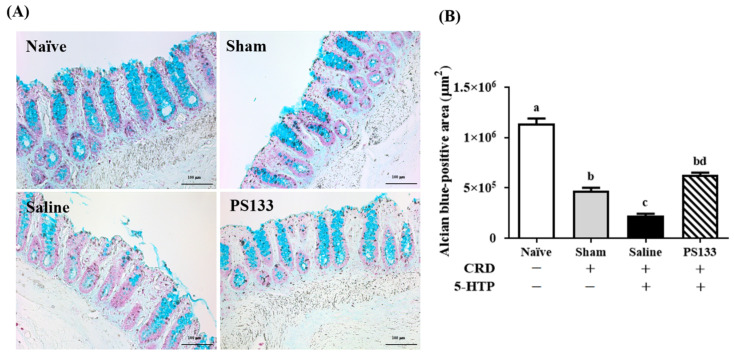
Oral administration of *L. cremoris* PS133 increased the amount of distal colonic mucin in an IBS-like rat model. (**A**) The mucins were stained with alcian blue at pH 2.5, and the blue-staining cells were shown in the field (scale bar = 100 μm). (**B**) Quantification of the alcian blue-positive area in each group was performed by the ImageJ software v1.42q. Data were expressed as mean ± S.E.M and analyzed by two-way ANOVA with Tukey’s post hoc test. Different letters (a, b, c, and d) above the bars indicate statistically significant differences at *p* < 0.05.

**Figure 4 ijms-26-02464-f004:**
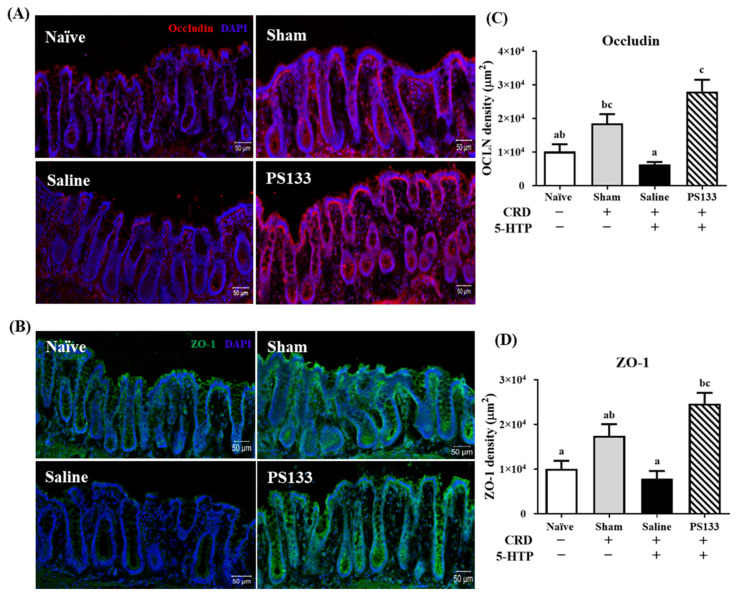
*L. cremoris* PS133 enhanced the gut barrier function. Representative immunofluorescence images of (**A**) occludin (red) and (**B**) ZO-1 (green) are shown in the distal colon tissue of the rats (scale bar in = 50 μm). Relative fluorescence intensities of (**C**) occludin and (**D**) ZO-1 were measured. Data were expressed as mean ± S.E.M and analyzed by two-way ANOVA with Tukey’s post hoc test. Different letters (a, b, and c) above the bars indicate statistically significant differences at *p* < 0.05.

**Figure 5 ijms-26-02464-f005:**
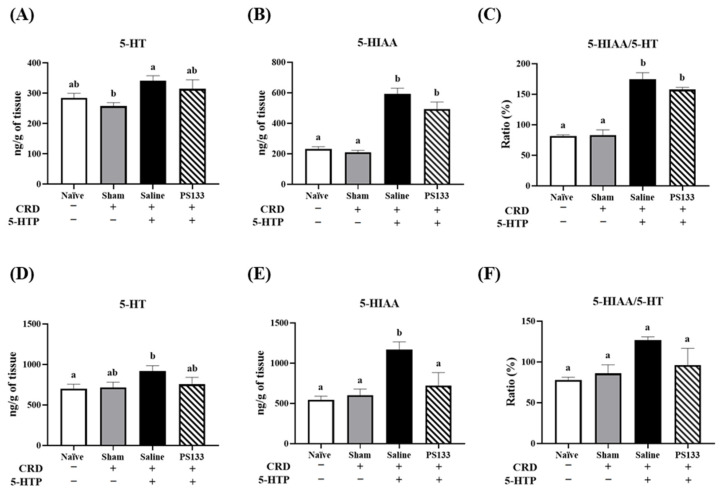
The levels of neurotransmitters in brain regions. The concentrations of 5-HT and 5-HIAA and the 5-HIAA/5-HT ratio in the prefrontal cortex (**A**–**C**) and hypothalamus (**D**–**F**) brain regions, respectively. Data were expressed as mean ± S.E.M and analyzed by two-way ANOVA with Tukey’s post hoc test. Different letters (a and b) above the bars indicate statistically significant differences at *p* < 0.05.

**Figure 6 ijms-26-02464-f006:**
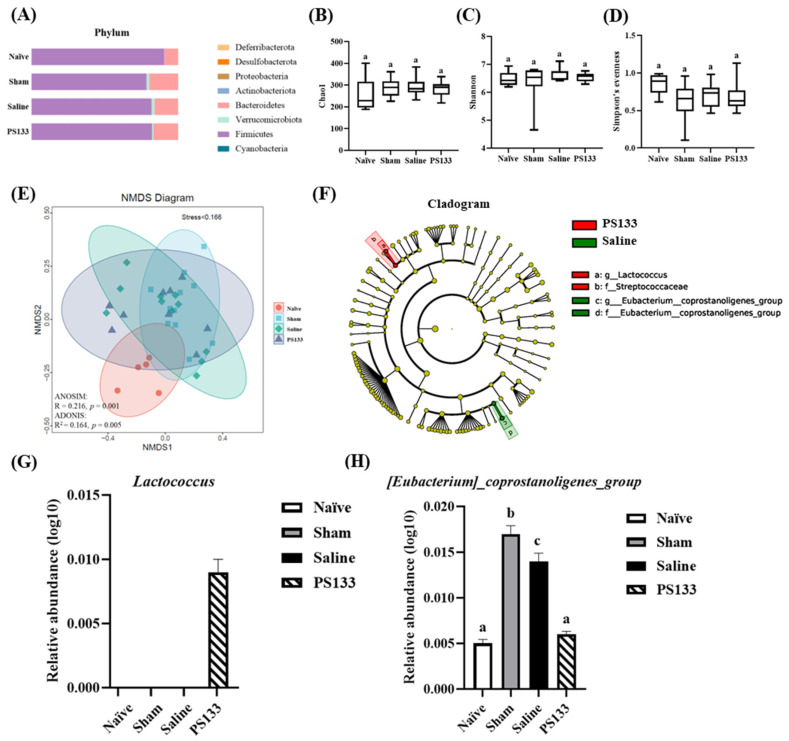
Effects of *L. cremoris* PS133 on gut microbiota composition. (**A**) Relative abundance of bacteria at the phylum level. Box plot showing the α-diversity in all groups: a comparison of bacterial diversity across sample groups in (**B**) the Chao1 index, (**C**) Shannon’s diversity index, and (**D**) Simpson’s evenness. Each bar represents the mean ± standard deviation. (**E**) A non-metric multidimensional scaling (NMDS) plot based on Bray–Curtis dissimilarity illustrating the β-diversity of gut microbiota among all groups. (**F**) LEfSe analysis of intestinal flora between the Saline group and PS133 group. Comparison of relative abundances of the (**G**) *Lactococcus* and (**H**) [*Eubacterium*]*_coprostanoligenes_group* level among all groups. Data were expressed as mean ± S.E.M and analyzed by one-way ANOVA with Tukey’s post hoc test. Different letters (a, b and c) above the bars indicate statistically significant differences at *p* < 0.05.

## Data Availability

The datasets supporting this article are available in the Appendix A. Further data are available from the corresponding author upon request.

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
