# Peer review of "Effects of *Lactococcus cremoris* PS133 in 5-Hydroxytryptophan-Induced Irritable Bowel Syndrome Model Rats"

_ijms, 2025, doi:10.3390/ijms26062464_

Round 1
Reviewer 1 Report
Comments and Suggestions for Authors
Very nice paper with interesting data and experimental methods. I have a few comments and suggestions for the authors.
1. Why do the sham surgery and naive rats have different levels of occludin, mucin, and zo-1? You would expect these two groups (naive and sham) have the same occludin and zo-1, correct? I know there was no statistically significant difference based on the statistical test (except mucin), but the authors should elaborate on this in then results section briefly and conclusion since the images and values look very different.
2. The authors use different statistical analyses between figures 2-5. I believe two-way ANOVA is the most appropriate test since you have the CRD and 5-HTP-treatment as variables. I would reanalyze the data to reflect this.
3. a. In Figure 6, there are no statistics are shown for the chao1 or shannon diversity indexes, please add these values even if not significant ('a' on all groups).
b. Is there a reason why peilou evenness wasn't shown?
c. Which beta diversity metric is being analyzed in figure 6D? Please add Bray Curtis to the figure legend.
d. ANOSIM is typically less sensitive than ADONIS, please show both ADONIS and ANOSIM statistics. It is better for transparency.
4. You should consider sequencing Lactococcus cremoris PS133 and designing primers specific to this strain or species if strain cannot be done reliably for future studies. It would be interesting to verify that the 16S results are due to bloom of your specific species or strain since genera can be any Lactococcus spp.
5. Additionally it would be nice to verify the isolation of Lactococcus cremoris PS133 by sequencing to confirm that only this specific strain was isolated.
6. It would be better to confirm viability of the Lactococcus cremoris PS133, to better understand whether it is beneficial as a mortibiotics or if it a gut-colonizing bacteria.
7. Correct spelling of Bray Curtis in the methods.
Reviewer 2 Report
Comments and Suggestions for Authors
This manuscript investigates the therapeutic effects of PS133 on a 5-HTP-induced IBS-like rat model. Overall, the writing is fluent, the experimental design is well-structured, and the data quality is excellent, with appropriate interpretation that aligns with the research hypothesis and objectives. Additionally, the manuscript thoroughly discusses the study's limitations and future directions. Therefore, only minor revisions are recommended for the manuscript.
Review recommendation: Research on the modulation of the microbiota-gut-brain axis (MGBA) by lactobacilli through 5-HTP to influence depression and anxiety differs from this study, which uses a 5-HTP-induced IBS-like animal model. However, the manuscript concludes, stating, “Our study findings suggest that L. cremoris PS133 improves IBS-like symptoms by modulating the MGBA.” This conclusion may be too strong. Unless the authors provide a more detailed explanation of how PS133 affects the regulation of 5-HIAA at the molecular level, a revision is recommended. This would aid in a better understanding of the probiotic’s therapeutic effects on IBS symptoms and establish a more specific connection to the microbiota-gut-brain axis (MGBA) theory.
